# *Mycobacterium avium* subsp. *paratuberculosis* MAP1889c Protein Induces Maturation of Dendritic Cells and Drives Th2-Biased Immune Responses

**DOI:** 10.3390/cells9040944

**Published:** 2020-04-11

**Authors:** Hye-Soo Park, Yong Woo Back, Yeo-Jin Son, Hwa-Jung Kim

**Affiliations:** Department of Microbiology and Department of Medical Science, College of Medicine, Chungnam National University, Daejeon 35015, Korea; 01027192188@hanmail.net (H.-S.P.); lenpk@nate.com (Y.W.B.); syj1234@cnu.ac.kr (Y.-J.S.)

**Keywords:** *Mycobacterium avium* subsp. *paratuberculosis*, MAP1889c protein, interleukin-10, dendritic cells

## Abstract

*Mycobacterium avium* subsp. *paratuberculosis* (MAP) is a causative agent of chronic granulomatous bowel disease in animals and is associated with various autoimmune diseases in humans including Crohn’s disease. A good understanding of the host-protective immune response and antibacterial immunity controlled by MAP and its components may contribute to the development of effective control strategies. MAP1889c was identified as a seroreactive antigen in Crohn’s disease patients. In this study, we investigated the immunological function of MAP1889c in dendritic cells (DCs). MAP1889c stimulated DCs to increase expression of co-stimulatory molecules (CD80 and CD86) and major histocompatibility complex (MHC) class molecules and to secret higher interleukin (IL)-10 and moderate IL-6, tumor necrosis factor (TNF)-α, and IL-12p70 levels through the Toll-like receptor (TLR) 4 pathway. MAP1889c-induced DC activation was mediated by mitogen-activated protein kinases (MAPKs), cAMPp-response element binding protein (CREB), and nuclear factor kappa B (NF-κB). In particular, the CREB signal was essential for MAP1889c-mediated IL-10 production but not TNF-α and IL-12p70. In addition, MAP1889c-matured DCs induced T cell proliferation and drove the Th2 response. Production of lipopolysaccharide (LPS)-mediated pro-inflammatory cytokines and anti-inflammatory cytokines was suppressed and enhanced respectively by MAP1889c pretreatment in DCs and T cells. Furthermore, treatment of MAP1889c in *M. avium*-infected macrophages promoted intracellular bacterial growth and IL-10 production. These findings suggest that MAP1889c modulates the host antimycobacterial response and may be a potential virulence factor during MAP infection.

## 1. Introduction

*Mycobacterium avium* subsp. *paratuberculosis* (MAP) is a pathogen that causes paratuberculosis or Johne’s disease (JD), which is a chronic granulomatous enteritis in ruminants [1,2]. MAP is of increasing interest because it can cause zoonosis through infected foods such as meat and dairy products. An association between MAP infection and human Crohn’s has been reported [3,4]. Similar to other mycobacterial strains, MAP can also survive and grow in mononuclear phagocytic cells, and it can develop a latent infection. Therefore, MAP and its components modulate the protective immune response of the host. However, little is known about the MAP components involved in the regulation of antibacterial immunity.

Immune responses with a dominant Th1 type have been observed during the early phase of paratuberculosis, with a shift to a dominant Th2 type with disease progression [5,6] induced by increased interleukin (IL)-10 [7,8]. It has been reported that MAP stimulates IL-10 secretion from ovine and bovine monocyte-derived macrophages [9,10] through activation of p38 mitogen-activated protein kinases (MAPKs) [11,12]. IL-10 is an anti-inflammatory cytokine which inhibits antimicrobial activity and the Th1 response [13] as well as increases the growth and persistent survival of MAP in macrophages by suppressing the production of pro-inflammatory cytokines [8]. It is well known that proteins and glycolipids of pathogenic mycobacteria are involved in regulating the production of pro- and anti-inflammatory cytokines in phagocytic cells. Mannosylated lipoarabinomannan (Man-LAM) derived from MAP induces rapid and prolonged production of IL-10 and facilitates the survival of MAP in macrophages [8,11]. Map41 of the MAP proline-proline-glutamic acid (PPE) protein family induces significant IL-10 as well as interferon (IFN)-γ production in peripheral blood mononuclear cells (PBMCs) from cattle infected with MAP [14,15]. Recently, six MAP recombinant proteins with a greater than 2-fold increase in IL-10 transcription in bovine macrophages have been reported [12]. However, little is known about MAP protein stimulation of IL-10 production in macrophages and/or dendritic cells (DCs) and the detailed underlying modulatory mechanism.

DCs are involved in the development of both the innate and adaptive immune system. Immature DCs are located in surrounding screened foreign antigens, including viral and microbial pathogens. During the uptake and processing of foreign antigens, immature DCs begin to mature and migrate to the spleen or adjacent lymph nodes. At maturity, DCs stimulate naïve T cells to differentiate into T cells that can produce anti- or pro-inflammatory immune responses, indicating that DCs play a critical role in determining the differentiation of Th1 or Th2 types, especially during mycobacterial infection including MAP. Several *Mycobacterium tuberculosis* (Mtb) proteins have been shown to induce DC maturation and to drive Th1 or Th2 responses [16,17]. Among MAP proteins, MAP1981c, a putative nucleic acid-binding protein, induces DC maturation and a Th1-biased response [18].

We identified MAP proteins that generate a strong IgG response in serum from patients with Crohn’s disease, and we analyzed their biological potential in DCs. Among them, we found that MAP1889c stimulated DCs to secrete higher levels of IL-10. MAP1889c, a conserved hypothetical protein, exhibits 86% homology of the protein sequence to Mtb Wag31 (Rv2145c), which plays a crucial role in cell division and cell wall synthesis [19], and it is associated with the cell surface and cell wall in the MAP K10 strain [20]. In this study, we investigated the activity of MAP1889c on DCs and the signaling pathway and functional role involved in MAP-1889c-mediated IL-10 production. Our data suggest that MAP1889c may act as a causal pathogenic factor underlying the upregulation of anti-inflammatory responses during MAP infection.

## 2. Materials and Methods

### 2.1. Ethics Statement

All animal experiments were performed in accordance with the Korean Food and Drug Administration (KFDA) guidelines for animal care and use. Animal work was done in accordance with procedures that were approved by the Institutional Animal Care and Use Committee of Chungnam National University, South Korea (Permit number: 201903A-CNU-5).

### 2.2. Mice

Specific pathogen-free, 5–6-week-old female C57BL/6 (H-2K^b^ and I-A^b^), C57BL/6J Toll-like receptor (TLR) 2 knockout (TLR2^-/-^; B6.129-Tlr2^tm1Kir^/J), C57BL/10 TLR4 knockout (TLR4^-/-^; C57BL/10ScNJ), and BALB/c mice were purchased from the Jackson Laboratory (Bar Harbor, ME, U.S.A) and used in all experiments. The mice were maintained under barrier conditions in a biohazard animal room at the Medical Research Center of Chungnam National University (Daejeon, Korea). The animals were fed a sterile commercial mouse diet with ad libitum access to water under standardized light-controlled conditions (12-h light and 12-h dark periods). The mice were monitored daily, and none of them showed any clinical symptoms or illness during this experiment.

### 2.3. Cells

Bone marrow-derived dendritic cells (BMDCs) and bone marrow-derived macrophages (BMDMs) were generated by flushing bone marrow cells from femurs and tibias. BMDCs were cultured for 6 days in Roswell Park Memorial Institute (RPMI) 1640 medium (Welgene Co., Daegu, Korea) supplemented with 10% fetal bovine serum (FBS) (Welgene), 100 unit/mL of penicillin/100 μg/mL of streptomycin (Welgene), 0.1 mM nonessential amino acids (Lonza, Basel, Switzerland), 50 μM β-mercaptoethanol (Lonza), 1 mM sodium pyruvate (Sigma-Aldrich, St. Louis, MO, USA), 20 ng/mL GM-CSF (CreaGene, Gyeonggi, Korea), and 10 ng/mL IL-4 (CreaGene). BMDMs were cultured for 6 days in Dulbecco’s modified eagle’s medium (Welgene) containing 10% FBS, 50 ng/mL mouse macrophage colony stimulating factor (M-CSF) (R&D System, Minneapolis, MN, USA), and 100 unit/mL of penicillin/100 μg/mL of streptomycin. Cultured cells were incubated at 37 °C in a 5% CO_2_ atmosphere.

### 2.4. Expression and Production of Recombinant Protein

To produce recombinant MAP1889c and Ag85B proteins, the corresponding gene was amplified by PCR using MAP ATCC 19698 or Mtb H37Rv ATCC 27294 genomic DNA as the template and the following primers respectively: *MAP1889c* forward 5′-CATATGCCGCCTACACCAGCCGACGTC-3′ and reverse 5′-AAGCTTAGTCAGTCGGAGCGGCTTCGC-3′; *Ag85B* forward 5′-GAATTCGATGACAGACGTGAGCCGAAAG-3′ and reverse 5′-AAGCTTGTTGTTGCCCCGGTTGAACTG-3. The PCR product of *MAP1889c* was cut with *Nde*I and *Hind*III, and *Ag85B* was cut with *EcoR*I and *Hind*III. The products were inserted into pET22b (+) vector (Novagen, Madison, WI, USA), and the resultants were sequenced. The recombinant plasmids containing *MAP1889c* and *Ag85B* were transformed into *E. coli* BL21 cells by heat shock for 1 min at 42 °C. The recombinant proteins were prepared as previously described [21].

### 2.5. Antibodies and Reagents

Endotoxin filter (END-X) and endotoxin removal resin (END-X B15) were purchased from the Associates of Cape Cod (East Falmouth, MA, USA). Fluorescein isothiocyanate (FITC)-annexin V/propidium iodide (PI) kits (556547) were acquired from BD Biosciences (San Jose, CA, USA). Lipopolysaccharides (LPS) from *Escherichia coli* (*E. coli*) O111:B4 (tlrl-eblps) and palmitoyl-3-Cys-Ser-(Lys)4 (Pam3CSK4, tlrl-pms) were purchased from InvivoGen (San Diego, CA, USA). Phycoerythrin (PE)-conjugated mAbs directed against CD80 (16-10A1), CD86 (GL1), major histocompatibility complex (MHC) class I (34-1-2S), MHC class II (I-A/I-E, M5/114.15.2), FITC-conjugated CD11c mAb (N418), BV605-conjugated CD4 mAb (563151), allophycocyanin (APC)-Cy^TM^7-conjugated CD8 mAb (557654), BV510-conjugated IL-10 mAb (563277), Alexa Fluor 488-conjugated T-bet mAb (561266), and PE-Cy^TM^7-conjugated GATA-3 mAb (560405) were purchased from eBioscience (San Diego, CA, USA). Mouse IL-10 (88-7105-77), IL-1β (88-7013-77), tumor necrosis factor (TNF)-α (88-7324-77), IL-6 (88-7064-77), IL-12p70 (88-7121-77), IL-4 (88-7044-77), IFN-γ (88-7314-77), IL-2 (88-7024-77), and IL-13 (88-7137-77) enzyme-linked immunosorbent assay (ELISA) kits were obtained from eBioscience. Dextran-FITC (molecular mass, 40,000 kDa) and poly-L-lysine hydrobromide (p2636) were acquired from Sigma. 4’,6-diamidino-2-phenylindole (DAPI) (D3571) and Texas Red^®^-X phalloidin (T7471) were obtained from Molecular Probes (Eugene, OR, USA). Alexa Fluor-488 goat anti-mouse IgG (A11001) and Alexa Fluor-488 goat anti-rabbit IgG (A11008) were purchased from Thermo Fisher Scientific (Waltham, MA, USA). Anti-TLR2 (H-175, sc-10739), anti-TLR4 (25, sc-293072), and anti-histidine (His, sc-8036) Abs were purchased from Santa Cruz Biotechnology (Paso Robles, CA, USA). Horseradish peroxidase-conjugated anti-mouse IgG (401215) and anti-rabbit IgG (401353) were acquired from Calbiochem (San Diego, CA, USA). Anti-phosphorylated extracellular signal-regulated kinases 1/2 (ERK)1/2 (T202/Y204, 9101), anti-phosphorylated p38 (T180/Y182, 9211), anti-phosphorylated IκB-α (S32, 14D4, 2859), anti-IκB-α (44D4, 4812), β-actin (13E5, 4970), anti-nuclear factor kappa B (NF-κB) p65 (D14E12, 8242), anti-phosphorylated cAMPp-response element binding protein (CREB) (S133, 87G3, 9198), and anti-CREB (48H2, 9197) Abs were obtained from Cell Signaling Technology (Danvers, MA, USA). Anti-T-bet (D-5), anti-GATA-3 (H-48), and anti-lamin B1 (H-90) were purchased from Santa Cruz Biotechnology. Specific inhibitors of ERK1/2 (U0126, 662005), p38 (SB203580, 559389), and NF-κB (BAY 11-7082, 196870) were purchased from Calbiochem, and CREB (666-15, 5661) was obtained from TOCRIS (Bristol, UK). Signal silence control siRNA (6568) and signal silence CREB PI3K siRNA I (6343) were purchased from Cell Signaling Technology. TurboFect transfection reagent was obtained from Thermo Fisher. Horseradish peroxidase-conjugated mouse IgG1 (0102-05) and IgG2a (0103-05) were purchased from SouthernBiotech (Birmingham, AL, USA).

### 2.6. Cytotoxicity Analysis

BMDCs were treated with MAP1889c (1, 5, and 10 μg/mL) or LPS (100 ng/mL) for 24 h. After incubation, the harvested BMDCs were stained with FITC-Annexin V and PI according to the manufacturer’s instructions. Data were collected on a FACSCanto flow cytometer (BD Biosciences) and analyzed using FlowJo data analysis software (Treestar, Inc., San Carlos, CA, USA).

### 2.7. Analysis of the Expression of Cell-Surface Molecules

BMDCs were treated with MAP1889c (1, 5, and 10 μg/mL) or LPS (100 ng/mL). After 24 h, cells were harvested and preincubated with 0.5% bovine serum albumin (BSA) in phosphate-buffered saline (PBS) for 30 min. The incubated cells were stained with anti-CD11c, anti-CD80, anti-CD86, anti-MHC-I, and anti-MHC-II Abs for 30 min at room temperature. The expression intensity of surface molecules was measured by flow cytometry (FACSCanto), and the data were analyzed using FlowJo data analysis software (Treestar).

### 2.8. ELISA for Cytokines

The culture supernatants were collected from BMDCs treated with each stimulant using various concentrations of MAP1889c or LPS. Sandwich ELISAs for detecting the cytokines in the culture supernatants were performed as recommended by the manufacturer (eBioscience). Plates were read on a Vmax kinetic microplate reader (Molecular Devices Co., Sunnyvale, CA, USA) at 450 mm. An amount less than the lower cutoff value for each cytokine was considered as zero. Calculating the standard curve for each cytokine, cutoff values of IL-10, IL-12p70, and IFN-γ were 62.5 pg/mL, and cutoff values of TNF-α and IL-6 were 31.25 pg/mL.

### 2.9. LPS Decontamination of Recombinant MAP1889c by Polymyxin B Binding

For pretreatment with polymyxin B (PmB, Sigma), LPS and MAP1889c were incubated in medium containing 50 μg/mL of PmB for 1 h at room temperature. After 24 h, cytokine levels in the supernatant of cells were analyzed by ELISA.

### 2.10. Analysis of Antigen Uptake Ability

MAP1889c or LPS treated-BMDCs were equilibrated at 37 °C or 4 °C for 30 min and then pulsed with FITC-conjugated dextran at a concentration of 1 mg/mL. Cold PBS was added to stop the reaction. The cells were washed and stained with anti-CD11c Ab, and then, CD11c^+^dextran^+^ cells were measured by flow cytometry.

### 2.11. Anti-MAP1889c Antibody

To obtain antiserum against MAP1889c, BALB/c mice were immunized intraperitoneally with 25 μg purified recombinant MAP1889c emulsified in incomplete Freund’s adjuvant (Sigma). Mice were injected with antigen three times at 2-week intervals, and the serum was collected 1 week after the final immunization.

### 2.12. Confocal Laser Scanning Microscopy

BMDCs were plated overnight in 12-well culture dishes containing poly-L-lysine-coated 18 mm-diameter round glass coverslips. The cells were then treated with MAP1889c for 1 h at 37 °C. After incubation, the cells were fixed in 4% paraformaldehyde, permeabilized in 0.1% Triton X-100, and then stained with anti-MAP1889c or anti-p65 Abs and imaged under a confocal microscope.

### 2.13. Immunoprecipitation

BMDCs were lysed with lysis buffer (50 mM Tris HCl, pH 8.0; 137 mM NaCl; 1 mM ethylenediaminetetraacetic acid (EDTA); 1% (*v*/*v*) Triton X-100; 10% (*v*/*v*) glycerol; 1 mM phenylmethylsulfonyl fluoride (PMSF); 1 μg/mL each of aprotinin, leupeptin, and pepstatin; 1 mM Na_3_VO_4_; 1 mM NaF; and proteinase inhibitor cocktail tablet (Roche, Basel, Switzerland)) for 20 min on ice. The cell lysate and 20 μg His-tagged MAP1889c protein were mixed and incubated at 4 °C for 6 h, and then His-tagged MAP1889c (His)-, TLR2-, and TLR4-associated proteins were immunoprecipitated by incubation with Ni-NTA Agarose (Qiagen, Hilden, Germany) or Dynabeads^®^Protein A (Thermo Fisher Scientific) for 24 h at 4 °C after incubation with an anti-mouse IgG Ab as a control Ab for anti-MAP1889c (His) and anti-rabbit IgG Ab as a control Ab for anti-TLR2 and anti-TLR4 for 4 h at 4 °C. The beads were collected, washed, and boiled in 5× sample buffer for 5 min. The bound proteins were analyzed by immunoblotting with anti-TLR2, anti-TLR4, and anti-His Abs.

### 2.14. Immunoblotting Analysis

After stimulation with 10 μg/mL MAP1889c, the cells were lysed with lysis buffer. Nuclear extracts from BMDCs were isolated using a subcellular protein fractionation kit for cultured cells (Thermo Fisher Scientific) according to the manufacturer’s instructions. Cell-lysate samples were resolved on SDS-polyacrylamide gels. Subsequently, the proteins were transferred onto a nitrocellulose membrane. The membranes were blocked in 5% skim milk and incubated with the Abs for 24 h at 4 °C, followed by incubation with HRP-conjugated secondary Abs for 1 h at room temperature. Epitopes on target proteins recognized specifically by Abs were visualized using the ECL advance kit (GE Healthcare, Little Chalfont, UK).

### 2.15. Cell Transfection

BMDCs were transfected with a control siRNA (siCON, 50 nM) or CREB siRNA (siCREB, 50 nM) using TurboFect for 24 h according to the manufacturer’s instructions. The transfection medium was then replaced with normal medium.

### 2.16. T Cell Proliferation Assay

For the T cell proliferation assay using the allogenic mixed lymphocyte reaction, BALB/c mice were immunized subcutaneously three times over a 2-week period with 50 μg of Ag85B or MAP1889c. BMDCs were treated for 24 h with MAP1889c (10 μg/mL) or LPS (100 ng/mL) and then pulsed with Ag85B (5 μg/mL). Splenic T cells were isolated from Ag85B-immunized mice, stained with 1 μM carboxyfluorescein succinimidyl ester (CFSE) for 10 min at 37 °C, and then co-cultured for 72 h with pretreated BMDCs. After 72 h of co-culture, the T cells were stained with anti-CD4 and CD8 Abs. CD4^+^CFSE^+^ and CD8^+^CFSE^+^ T cells were analyzed by flow cytometry.

### 2.17. Intracellular Staining in T Cells

For protein immunization, C57BL/6 mice were immunized with Ag85B or MAP1889c (50 μg/mice) via intravenous injection administered three times at 2-week intervals. Two weeks after the final immunization, the spleens were isolated from the mice. The single-cell suspensions were then filtered through a 40-μm cell nylon mesh cell strainer, treated with red blood cell lysis buffer (Sigma) for 5 min, and washed twice with RPMI 1640 medium supplemented with 2% FBS. Single-cell suspensions of the spleen of immunized mice were stimulated with Ag85B (10 μg/mL) or MAP1889c (10 μg/mL) for 12 h at 37 °C in the presence of GolgiStop (BD Biosciences). The cells were first blocked with Fc Block (anti-CD16/32; eBioscience) for 15 min at 4 °C and then stained with fluorochrome-conjugated anti-CD4 or anti-CD8 Ab (BD Biosciences) for 30 min at 4 °C. Cells stained with the appropriate isotype-matched immunoglobulins were used as a negative control. The cells were fixed and permeabilized using a Cytofix/Cytoperm kit (BD Biosciences) according to the manufacturer’s instructions. Intracellular anti-IL-10, anti-T-bet, or anti-GATA-3 (BD Biosciences) levels were detected with fluorescein-conjugated antibodies in permeation buffer. Then, the samples were detected by flow cytometry.

### 2.18. Analysis of Bacterial Growth

BMDMs were infected with *Mycobacterium avium* (strain 104) at a multiplicity of infection (MOI) of 1 for 4 h at 37 °C, 5% CO_2_. Amikacin (200 μg/mL; Sigma) was added to each well, and the cells were incubated for 2 h to kill extracellular bacteria, washed three times with PBS, and then treated with MAP1889c or LPS for 72 h. After incubation, the cells were lysed with sterile distilled water for 30 min. The lysates were serially diluted and plated onto 7H10 agar plates to determine the “input” bacterial numbers.

### 2.19. Statistical Analysis

All experiments were repeated at least 3 times with consistent results. The levels of significance for comparisons between samples were determined by Tukey’s multiple comparison test distribution or two-way ANOVA using statistical software (GraphPad Prism Software, version 4.03; GraphPad Software, San Diego, CA). The data in the graphs are expressed as the mean values ± SD; * *p* < 0.05, ** *p* < 0.01, or *** *p* < 0.001 were considered statistically significant.

## 3. Results

### 3.1. Purification of MAP1889c Protein

A recombinant MAP1889c protein purified from *Escherichia coli* extracts was observed as a major band at 35 kDa by SDS-PAGE and strongly reacted with anti-His antibodies (Figure 1A). The purified protein did not show any cytotoxic effects on bone marrow-derived dendritic cells (BMDCs) at the tested concentration (Figure 1B).

### 3.2. MAP1889c Induces DC Maturation Accompanied by Higher IL-10 Production

We investigated whether MAP1889c could induce DC maturation. Because the antigen uptake capability of DCs decreases as DC mature, we tested antigen uptake ability during DC maturation. As shown in Figure 2, uptake of FITC-dextran was significantly reduced in LPS- or MAP1889c-treated BMDCs at 37 °C when compared with untreated BMDCs. At concentration of 10 μg/mL, MAP1889c-induced DC maturation was comparable to that with LPS (100 ng/mL). There was no endocytic activity at 4 °C.

Next, phenotypic maturation markers of DCs such as co-stimulatory and MHC-class molecules were analyzed by flow cytometry. LPS was used as a positive control. As shown in Figure 3A, the expressions of CD80 and CD86 as well as MHC class I and II molecules in BMDCs treated with MAP1889c were significantly increased in a dose dependent manner when compared to the untreated control. We simultaneously assessed the inflammatory cytokines in the culture supernatant. As expected, MAP1889c induced significant production of IL-10, IL-1β, IL-6, and IL-12p70 in a dose-dependent manner when compared with untreated cells (Figure 3B). Production of TNF-α was significantly higher in MAP1889c-treated than untreated cells, but it did not show significant dose-dependent increase. Interestingly, other pro-inflammatory cytokines except IL-1β were significantly higher in LPS (100 ng/mL)-stimulated DCs compared with MAP1889c-treated cells at concentration of 10 μg/mL, but IL-10 production was significantly higher in MAP1889c- compared with LPS-treated cells. We next confirmed that MAP1889c-mediated cytokine production was not due to LPS contamination, although LPS was removed from the purified MAP1889c protein. MAP1889c-mediated IL-10, TNF-α, and IL-12p70 production was not affected by polymyxin B (PmB) pretreatment, which completely abrogated LPS-induced cytokine production (Appendix A). Taken together, these data suggest that MAP1889c can induce DC maturation and activation with elevated IL-10 and moderate IL-6, TNF- α, and IL-12p70 production.

### 3.3. MAP1889c Induces DC Activation Through TLR4 Interaction

We determined the involvement of TLR signaling in MAP1889c-mediated DC activation by using the BMDCs prepared from C57BL/6 wild-type (WT), TLR2^-/-^, and TLR4^-/-^ mice. Confocal microscopy showed that MAP1889c interacted with the surface of BMDCs from WT and TLR2^-/-^ mice but not TLR4^-/-^ mice (Figure 4A). Immunoprecipitation analysis using anti-His or anti-TLR2 and anti-TLR4 Abs also revealed that MAP1889c interacted with TLR4 but not TLR2 (Figure 4B). Next, cytokine production and surface molecule expression in BMDCs from C57BL/6 wild-type (WT), TLR2^-/-^, and TLR4^-/-^ mice after MAP1889c treatment were determined. As expected, LPS and Pam3CSK4 activities were significantly decreased in BMDCs from TLR4^-/-^ and TLR2^-/-^-mice, respectively. MAP1889c-mediated cytokine production and expression of surface molecules were significantly depressed in BMDCs from TLR4^-/-^ mice compared with WT mice or TLR2^-/-^-mice (Figure 4C,D). These results suggest that the TLR4 pathway is involved in MAP1889c-induced DC activation.

### 3.4. MAP1889c Induces DC Activation through MAPK, and NF-κB Pathways

We determined whether mitogen-activated protein kinases (MAPKs) and nuclear factor kappa B (NF-κB) signaling pathways were involved in MAP1889c-mediated DC activation. As shown in Figure 5A, MAP1889c triggered the phosphorylation of ERK1/2 and p38 as well as the phosphorylation and degradation of IκB-α in BMDCs. Significant translocation of p65 from the cytosol to the nucleus was also observed (Figure 5B). To confirm the role of MAPKs in MAP1889c-induced cytokine production and the expression of surface molecules, BMDCs were pretreated with ERK1/2 inhibitor (U0126), p38 inhibitor (SB203580), and NF-κB inhibitor (Bay11-7082) prior to MAP1889c treatment. As shown in Figure 5C, IL-10, TNF-α, and IL-12p70 production was significantly inhibited by the MAPK and NF-κB inhibitors. MAP1889c-mediated expression of CD80 and MHC class I was also significantly inhibited by pretreatment with MAPKs and NF-κB inhibitors (Figure 5D).

### 3.5. CREB Signals Are Essential for MAP1889c-Mediated IL-10 Production

As indicated in Figure 3B, MAP1889c induced significantly higher IL-10 production. Because CREB is known to play an essential role in IL-10 production in immune cells, we examined whether MAP1889c could induce IL-10 production via CREB signaling pathways. We found that MAP1889c induced the phosphorylation of CREB (Figure 5E).

MAP1889c-induced IL-10 production but not TNF-α and IL-12p70 production was significantly inhibited by BMDC pretreated with CREB inhibitor (666-15) (Figure 5C), which also suppressed the expression of CD80 and MHC-class I induced by MAP1889c (Figure 5D). Furthermore, BMDCs transfected with siCREB showed suppression of MAP1889c-mediated CREB phosphorylation (Figure 5F) and significant inhibition of MAP1889c-mediated IL-10 production but not TNF-α or IL-12p70 production (Figure 5G). Our data demonstrate that the CREB signaling pathway is essential for IL-10 production induced by MAP1889c in BMDCs.

### 3.6. MAP1889c-Matured DCs Induce T Cell Proliferation and Th2 Responses

We investigated whether MAP1889c-treated DCs could induce T cells proliferation. To achieve this goal, we performed allogenic mixed lymphocyte reaction (MLR) assays using antigen-specific T cells from Ag85B-immunized BALB/c mice. DCs were derived from C57BL/6 mice. LPS- and MAP1889c-treated DCs elicited significant proliferation of CD4^+^ and CD8^+^ T cells compared with nontreated DCs (Figure 6A). A similar reactive pattern was observed in the syngeneic MLR assay, but no significant difference was observed among DCs treated with each Ag (Figure 6B). GATA-3 transcription factor expression, and IL-10 and IL-4 production were associated with Th2 cell differentiation. For these reasons, we determined the transcription factor expression and cytokine production when Ag-matured DCs were co-cultured with syngeneic naïve splenocytes. As shown in Figure 6C, the expression of GATA-3 in splenocytes activated by MAP1889c-matured DCs was increased but T-bet expression in splenocytes was enhanced by LPS-matured DCs. In addition, the productions of IL-10 and IL-4 were significantly higher in splenocytes activated by MAP1889c-matured DCs compared with untreated or LPS-matured DCs. In contrast, the production of cytokines involved in the Th1 response, IFN-γ, IL-12p70, and IL-2 was significantly lower in splenocytes activated by MAP1889c-matured DCs compared with those by LPS-matured DCs (Figure 6D). Further, the frequency of IL-10-producing CD4^+^ and CD8^+^ T cells was significantly increased during co-culture with MAP1889c-matured DCs compared with untreated or LPS-treated DCs (Figure 6E).

Next, we evaluated whether MAP1889c could generate Th2 immune responses ex vivo. Splenocytes from Ag85B- or MAP1889c-immunized mice were restimulated with the same antigen used for immunization. Western blot analysis showed that T-bet expression was increased in splenocytes stimulated with Ag85B and that GATA-3 expression was increased by MAP1889c-specific stimulation in splenocytes (Figure 7A). The frequencies of GATA-3^+^ or T-bet^+^ T cells in splenocytes stimulated specifically with each Ag were analyzed by flow cytometry. As shown in Figure 7B, the frequencies of T-bet^+^ in CD4^+^ and T-bet^+^ CD8^+^ T cells were reduced in response to MAP1889c-specific stimulation compared with Ag85B-specific stimulation. However, a significant difference in GATA-3^+^ CD4^+^ and GATA-3^+^ CD8^+^ T cells was not detected between the MAP1889c- and Ag85B-specific stimulated splenocytes groups. Further, IFN-γ levels were significantly higher in the culture supernatant of Ag85B-specific than MAP1889c-specific stimulated splenocytes, but there was no significant difference in the IL-2 level and no detectable IL-4 (Figure 7C). Taken together, these data indicate that MAP1889c-matured DCs induce T cell proliferation and show a biased response to Th2.

### 3.7. MAP1889c Suppresses the LPS-Induced Pro-Inflammatory Response and Promotes Intracellular Bacterial Survival

As previously described, MAP1889c-matured DCs induced a prominent Th2 response in vitro (Figure 6), but the ex vivo results (Figure 7) suggest that MAP1889c induces suppression of the Th1 response and no enhancement of the Th2 response. Th1 responses are diminished in the presence of IL-10 during Mtb infection [22]. In addition, mice lacking Th1-associated cytokines (IL-12, or IFN-γ) and transcription factors (T-bet) are highly susceptible to Mtb infection [23]. Therefore, we hypothesized that MAP1889c can suppress Th1 response via IL-10 production. To test this possibility, we tested whether MAP1889c could affect the activity of LPS on DCs. Although a similar cytokine-producing pattern between LPS- or MAP1889c-stimulated DCs was observed (Figure 3B), BMDCs pretreated with MAP1889c showed significant inhibition of LPS-mediated TNF-α and IL-12p70 production as well as a significant increase in LPS-mediated IL-10 production when compared to DCs treated with LPS alone (Figure 8A). The same experiment was then performed in a co-culture system with DCs and T cells. As shown in Figure 8B, T cells activated by LPS-stimulated DCs pretreated with MAP1889c showed significantly lower production of IFN-γ, TNF-α, and IL-12p70 but higher production of IL-10 and IL-13 when compared to LPS alone-treated DCs. These results indicate that MAP1889c downregulates pro-inflammatory and Th1 responses and upregulates anti-inflammatory responses in DCs.

Based on our previous results, we could predict the potential of MAP1889c as a virulence factor in MAP. Therefore, we investigated the role of MAP1889c in bacterial survival. To achieve this goal, bone marrow-derived macrophages (BMDMs) were infected with *Mycobacterium avium* and then stimulated with MAP1889c or LPS. A significant increase in mycobacterial growth was observed in MAP1889c-stimulated BMDMs compared with nontreated infected cells or cells stimulated with LPS (Figure 8C). IL-10 production in MAP1889c-stimulated BMDMs was significantly higher than in nontreated infected cells or cells stimulated with LPS, while the productions of TNF-α and IL-12p70 were significantly lower in MAP1889c-treated cells (Figure 8D). A similar cytokine-producing pattern was observed in uninfected BMDMs (Appendix A). These results suggest that MAP1889c might plays a role as a virulence factor via IL-10 production during bacterial infection.

## 4. Discussion

The expression of anti-inflammatory cytokines, including TGF-β and IL-10, is higher in tissue lesion of cows that have progressed to the clinical stage of paratuberculosis compared with subclinically infected or healthy cows [24], suggesting a role for IL-10 in disease progression. It is important to identify and characterize the proteins that stimulate phagocytic cells to secrete IL-10 production for the development of new control strategies against MAP infection. Here, we report a newly identified MAP1889c that induced a dominant Th2 response in vitro and further suppressed the Th1 response without enhancement of the Th2 response ex vivo, which was related to higher MAP1889c-mediated IL-10 production. In the present study, we support a role for IL-10 induced by MAP protein through immune networks during bacterial infection.

The discovery of novel mycobacterial antigens and their role in host immunity can contribute to the development of effective defense strategies, including vaccines and drug targets. Several MAP proteins have been studied for their immunological roles in various immune cell-mediated responses [25]; recombinant MAP Ag85 induced not only significant lymphocyte proliferation but also production of IFN-γ, TNF-α, IL-2, and IL-12 but not IL-4 in PBMCs from MAP-infected cows. In a preliminary study, we identified cell extract proteins of MAP that reacted strongly with the sera of Crohn’s disease patients. MAP1889c was one of the proteins that strongly reacted with the sera of the patients. *M. tuberculosis* Wag31 or antigen 84 with homology to MAP1889c was also first identified as a seroreactive antigen in tuberculosis and leprosy patients [26], and it plays a critical role in cell wall synthesis [19]. It was recently reported that Wag31 inhibits anti-CD3 and anti-CD28-mediated IL-10 and IL-17 production in human T cells [27], but this study was performed in purified T cells without antigen presenting cell (APCs) involvement. In this study, we investigated biological activity of MAP1889c in DCs, which play critical roles in the differentiation of naïve T cells into specific immune types [28]. We found that MAP1889c induced DC activation, which showed significantly higher IL-10 production but lower IL-6, TNF-α, and IL-12 production compared with LPS-treated DCs (Figure 3B). Further, MAP1889c-matured DCs induced T cell proliferation and drove a Th2-biased immune response (Figure 6). Mtb PE25/PPE41 induces DC maturation with higher IL-10 and TNF-α but lower IL-12 production compared with LPS, and PE25/PPE41complex-treated DCs promote Th2 polarization [29]. Another report has also demonstrated that Rv1917c-treated DCs secrete high levels of IL-6, IL-8, and TNF-α as well as IL-10 but not IL-12 and stimulate CD4^+^ T cells to produce Th2 cytokines [30]. It has been reported that several MAP proteins, including MAP1981c, MAP2541c (malate dehydrogenase), MAPCobT, and MAP1305, also induce DC maturation and Th1 polarization [18,31,32,33]. However, no reports have examined MAP proteins that induce Th2 polarization through DC maturation.

Various immune cells, including macrophages and DCs, have multiple pathogen recognition receptors (PRRs). TLRs are a family of cell membrane receptors that initiate cellular signal transduction associated with mycobacterial infection [34]. We demonstrated that MAP1889c binds TLR4 molecules in DCs. Other MAP proteins, such as MAP1305, CobT, and MAP1989c, induce DC maturation via the TLR4 signaling pathway [32,33]. MAP stimulates bovine mononuclear cells to secrete higher IL-10 production via activation of MAPK-p38 and interactions with TLR2 [11,35]. MAP1889c-medicated DC activation and cytokine production are involved in the MAPK and NF-κB pathways. In particular, the CREB signal is essential for MAP1889c-mediated IL-10 production but not TNF-α and IL-12 production in BMDCs. The signaling pathways responsible for the production and regulation of IL-10 in immune cells are diverse, mainly including MAPK, NF-κB, signal transducer and activator of transcription 3 (STAT3), and CREB. Among them, CREB, induced by various growth factors and inflammatory signals, plays an essential role in IL-10 production in immune cells [36,37]. Our data also support involvement of the CREB signal in MAP-mediated IL-10 production.

Naïve T cells differentiate into subpopulation of activated effector Th cells when bound to antigen presented by APCs [38]. Activated T cells are critical for the initiation and regulation of the immune response for defense against intra- and extracellular pathogens. T cells that differentiate into the Th1 type secrete IFN-γ and TNF-α and activate macrophages to kill microorganisms located within phagocytic cells. Conversely, Th2 cells secrete IL-4, 5, 10, and 13 and primarily defend against extracellular pathogens [39,40]. In the present study, MAP1889c-treated DCs also induced T cell proliferation with higher IL-10 and IL-4 production, expansion of IL-10-producing T cells, and expression of GATA-3 in MLR assays (Figure 6 and Figure 7). In many studies, naïve T cell proliferation experiments with mature DC are typically performed using an ovalbumin (OVA)-specific transgenic mouse model [16,41]. This mouse model was not available in our laboratory. Instead, we observed that MAP1889c-matured DC could induce T cell proliferation when antigen-activated DCs were co-cultured with splenic T cells from Ag85B-immunized mice followed by the addition of Ag85B to the culture media (Figure 6) as described previously [42]. We confirmed that MAP1889c-matured DCs could induce Th2 responses in vitro. However, when splenocytes from Ag85B- or MAP1889c-immunized mice were restimulated with the same antigen that was used for immunization, MAP1889c induced suppression of the Th1 response without enhancement of the Th2 response (Figure 7). We further found that pretreatment of MAP1889c suppressed the LPS-mediated Th1 type response (Figure 8A,B). Mtb PPE18 activates IL-10 induction by interacting with TLR2 in macrophages, and it also inhibits LPS-induced IL-12p40 production in macrophages [43]. In fact, although IL-10 is known to be a cytokine that is closely related to Th2 immune responses, the mechanisms of Th2 initiation and development associated with Th2-DCs are poorly understood [44,45]. In addition, little research has investigated DC-secreted IL-10 induction of Th2 immune responses. At present, more detailed mechanisms related to MAP1889c-induced Th2 polarization via DC maturation are under investigation.

IL-10 has been shown to interfere with Th1 cell and macrophage function, and IL-10 deficiency improves the outcome of Mtb infection, mainly due to enhanced macrophage and Th1 responses [46,47]. Therefore, hypersecretion of IL-10 provides a niche for the continued survival of pathogens in vitro and in vivo [11,48]. Therefore, we finally determined the effect of MAP1889c on the intracellular survival of *M. avium*, which was used because of its ease of experimentation instead of MAP as previously described [12]. Treatment of MAP1889c in *M. avium*-infected macrophages promoted intracellular bacterial growth and IL-10 production compared with LPS treatment (Figure 8C,D). It has been reported that MAP-derived Man-LAM suppresses killing of *M. avium* in bovine macrophages by regulating phagosome maturation, but its suppression effect was not affected by a neutralizing anti-IL-10 antibody [11]. Another paper also reported that MAP proteins that induce IL-10 expression significantly enhance the phosphorylation of MAPK-p38, but there is no correlation between their capability in IL-10 expression and inhibition of killing of *M. avium* in macrophages [12]. There are some differences in experimental methods compared with our study. In both papers, Man-LAM and the proteins were treated in macrophages before bacterial infection but MAP1889c was treated after infection. Although the amount of IL-10 produced by LAM or other proteins could not be directly compared with MAP1889c, our data suggest that, when compared to LPS, MAP1889c may stimulate the cells to secrete IL-10 in sufficient quantities to enhance intracellular *M. avium* growth. Taken together, we predict that MAP1889c is a potential virulence factor during MAP infection, although MAP was not used to evaluate intracellular survival.

## Figures and Tables

**Figure 1 cells-09-00944-f001:**
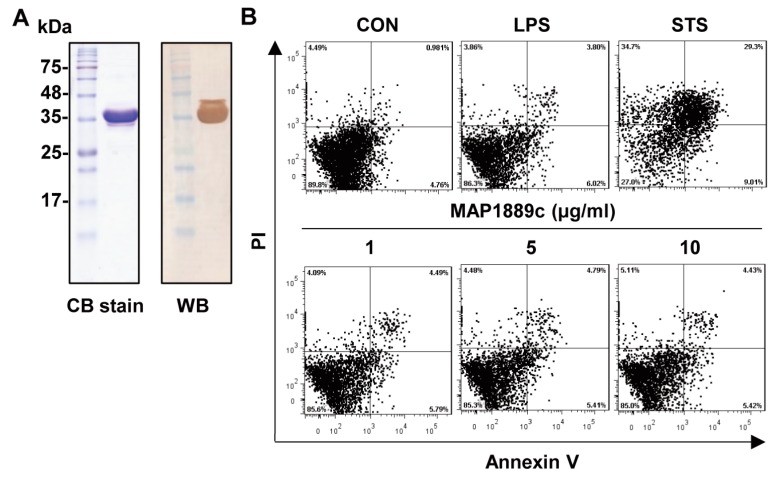
Preparation of the MAP1889c protein: (**A**) MAP1889c purified from *E. coli* extracts was subjected to SDS-PAGE and Western blot analysis using a mouse anti-His antibody. All proteins were analyzed by SDS-PAGE with Coomassie blue staining. (**B**) The cytotoxic effects of MAP1889c were analyzed by flow cytometry. Bone marrow-derived dendritic cells (BMDCs) were stimulated with MAP1889c (1 to 10 μg/mL), lipopolysaccharide (LPS) (100 ng/mL), or staurosporine (STS, 100 nM) for 24 h and then stained with Annexin V and propidium iodide (PI). The percentage of positive cells in each quadrant is indicated. The results are representative of three experiments.

**Figure 2 cells-09-00944-f002:**
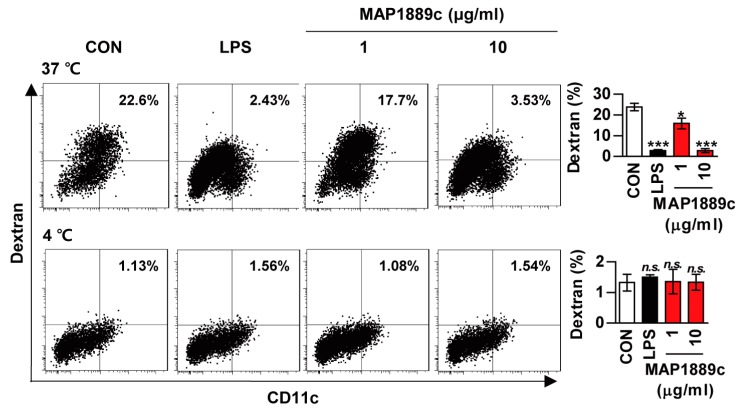
Endocytic activity of MAP1889c-treated dendritic cells (DCs): Bone marrow-derived dendritic cells (BMDCs) were treated with 100 ng/mL Lipopolysaccharide (LPS) or 1 or 10 μg/mL MAP1889c for 24 h, incubated with dextran- fluorescein isothiocyanate (FITC) at 37 °C or 4 °C for 1 h, and then stained with a phycoerythrin (PE)-conjugated anti-CD11c^+^ antibody. Endocytic activity was assessed by flow cytometric analysis of dextran-FITC uptake. The percentage of CD11c^+^dextran^+^ cells is indicated. The bar graphs depict the mean values ± SD (n = 3). * *p* < 0.05 and *** *p* < 0.001 for treatment compared with the untreated controls (CON) or for the difference between treatment data. *n.s*., no significant difference.

**Figure 3 cells-09-00944-f003:**
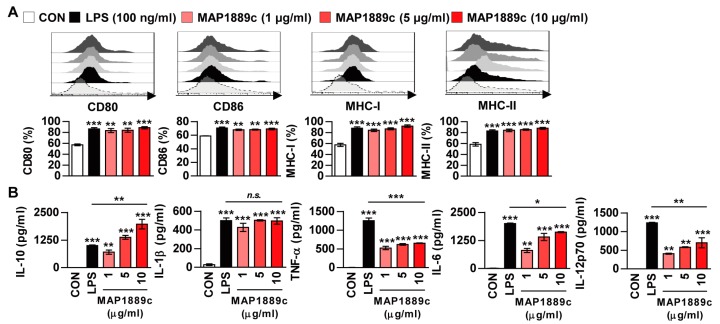
MAP1889c induces BMDC maturation and high levels of IL-10 production. The BMDCs were stimulated with MAP1889c (1, 5, or 10 μg/mL) and LPS, 100 ng/mL for 24 h. (**A**) The expressions of surface markers were analyzed by two-color flow cytometry. The cells were gated to exclude CD11c^+^ cells. BMDCs were stained with anti-CD80, anti-CD86, anti-major histocompatibility complex (MHC) class I, or anti-MHC class II antibodies. The histograms are representative of five experiments. The bar graphs show the percentage (mean ± SD of five experiments) for each surface molecule on CD11c^+^ cells. (**B**) Interleukin (IL)-10, IL-1β, tumor necrosis factor (TNF)-α, IL-6, and IL-12p70 cytokines from the culture supernatants were measured by enzyme-linked immunosorbent assay (ELISA). All data are expressed as the mean ± SD (*n* = 3). All data are expressed as the mean ± SD (*n* = 3). * *p* < 0.05, ** *p* < 0.01, and *** *p* < 0.001 for treatment compared with untreated controls (CON) or for the difference between treatment data.

**Figure 4 cells-09-00944-f004:**
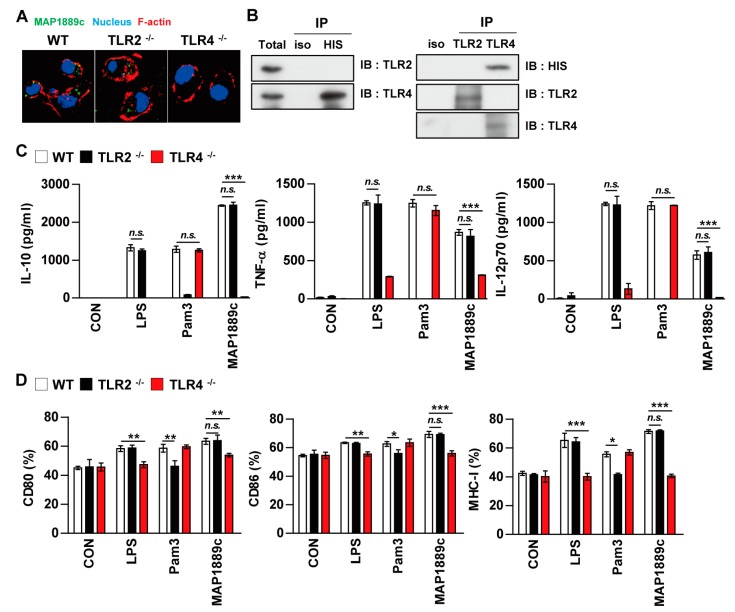
MAP1889c induces DC activation via Toll-like receptor (TLR) 4 pathways. BMDCs derived from wild-type (WT), TLR2^–/–^, and TLR4^–/–^ mice were treated with MAP1889c (10 μg/mL), LPS (100 ng/mL), or Pam3CSK4 (Pam3) (100 ng/mL) for 24 h. (**A**) MAP1889c-treated BMDCs for 1 h were fixed and then stained with 4’,6-diamidino-2-phenylindole (DAPI) (blue) and Alexa Fluor-488-conjugated anti-MAP1889c antibody. Representative images out of three independent experiments are shown. Scale bar, 10 μm. (**B**) The cell lysates from BMDCs treated with MAP1889c for 6 h were used for immunoprecipitation with anti-mouse IgG and anti-His or with anti-TLR2 and anti-TLR4 antibodies. Thereafter, proteins were detected using immunoblotting with anti-His, anti-TLR2, or anti-TLR4 antibodies. The total is shown as the mean total cell lysates (input). (**C**) Production of IL-10, TNF-α, and IL-12p70 in the culture supernatants was determined by ELISA. All data are expressed as the mean ± SD (*n* = 3). (**D**) Expression of CD80, CD86, and MHC class I molecules on BMDCs stimulated with each antigen was determined by staining and flow cytometry. The bar graphs show the mean percentage ± SEM of each surface molecule on CD11c^+^ cells across three independent experiments. * *p* < 0.05, ** *p* < 0.01, and *** *p* < 0.001 for treatment values in BMDCs from TLR2^–/–^ or TLR4^–/–^ mice compared with MAP1889c-, LPS-, or Pam3CSK4-treated BMDCs from WT mice.

**Figure 5 cells-09-00944-f005:**
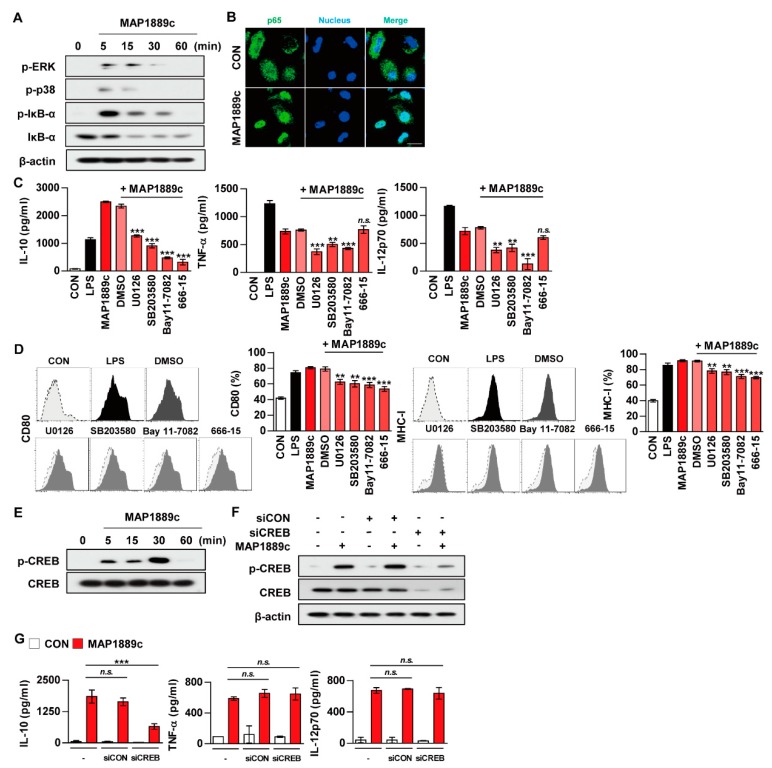
MAP1889c induces DC maturation via mitogen-activated protein kinases (MAPK), nuclear factor kappa B (NF-κB), and cAMPp-response element binding protein (CREB) pathways, and the CREB signal is involved in IL-10 production. BMDCs stimulated with MAP1889c for the indicated times were lysed, and the proteins in the total cell lysate were separated by SDS-PAGE followed by immunoblot analysis using antibodies against (**A**) phospho-extracellular signal-regulated kinases (ERK)1/2, phospho-p38, phospho-IκB-α, IκB-α, and β-actin. This image is representative of three experiments showing similar results. (**B**) BMDCs were plated in covered glass chamber slides and treated with MAP1889c for 1 h, and the immunoreactivity of the p65 subunit of NF-κB in cells was determined by immunofluorescence. Scale bar, 10 μm. (**C**,**D**) BMDCs were pretreated with pharmacological inhibitors of ERK (U0126, 10 μM), p38 (SB203580, 20 μM), NF-κB (BAY11-7082, 5 μM), CREB (666-15, 5 μM), or dimethyl sulfoxide (DMSO, Sigma) (vehicle control) for 1 h prior to treatment with MAP1889c (10 μg/mL). After 24 h, the amounts of IL-10, TNF-α, and IL-12p70 in the culture medium were measured by ELISA (**C**). The mean ± SD is shown for three independent experiments. The expression levels of CD80 and MHC-I were analyzed by flow cytometry (**D**). Bar graphs show percentages (mean ± SD of three separate experiments) for each surface molecule on CD11c^+^ cells. ** *p* < 0.01 or *** *p* < 0.001 for each inhibitor treatment compared with MAP1889c-treated controls. (**E**) Total lysates of BMDCs stimulated with MAP1889c for the indicated times were separated by SDS-PAGE, followed by immunoblot analysis using phospho-CREB and CREB. (**F**,**G**) BMDCs were transfected with CREB siRNA (siCREB) or nonspecific siRNA as a control (siCON) and then with MAP1889c (10 μg/mL) for 30 min (**F**) or 24 h (**G**). (**F**) Immunoblot analysis using antibodies against phospho-CREB and CREB. (**G**) IL-10, TNF-α, and IL-12p70 production in the culture supernatant were measured by ELISA. All data are expressed as the mean ± SD (*n* = 3). All data are expressed as the mean ± SD (*n* = 3). *** *p* < 0.001 for siRNA transfection compared with MAP1889c-treated cells.

**Figure 6 cells-09-00944-f006:**
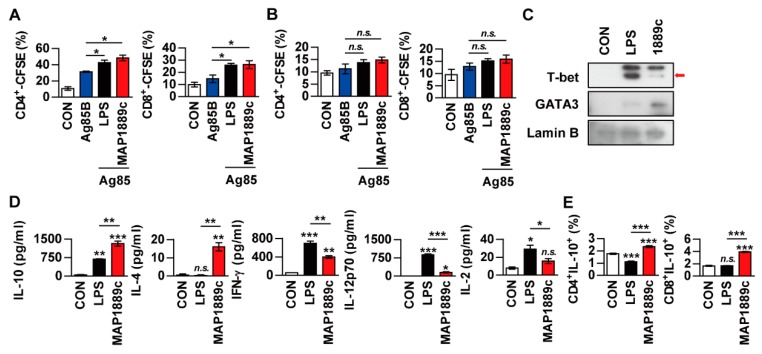
MAP1889c-activated BMDCs induce T cell proliferation and a Th2 response. (**A**,**B**) BMDCs from C57BL/c were activated with MAP1889c (10 ug/mL) or LPS- (100 ng/mL), pulsed with Ag85B (10 ug/mL) for 24 h, and then co-cultured with carboxyfluorescein succinimidyl ester (CFSE)-stained splenocytes isolated from Ag85B-immunized BALB/c (**A**) or C57BL/6 (**B**) mice for 72 h. The proliferation of CD4^+^ and CD8^+^ T cells was then assessed by flow cytometry. CON, T cells co-cultured with untreated DCs. (C–E) Unstimulated, LPS-stimulated, and MAP1889c-stimulated DCs were co-cultured with naïve splenocytes of C57BL/6 mice at a DC to splenocyte ratio of 1:10. After 72 h, the levels of T-bet and GATA-3 expression in the T cells were assessed by immunoblotting (**C**). The cytokine levels in the culture supernatants were measured by ELISA (**D**). Subsequently, IL-10-producing CD4^+^ T cells (CD4^+^IL-10^+^) and IL-10-producing CD8^+^ T cells (CD8^+^IL-10^+^) were gated as shown (**E**). Gating strategies are shown in Appendix A. All data are expressed as the mean ± SD (*n* = 3). * *p* < 0.05, ** *p* < 0.01, and *** *p* < 0.001 for the treatment compared with untreated controls (CON) or for the difference between treatment data. *n.s*., no significant difference.

**Figure 7 cells-09-00944-f007:**
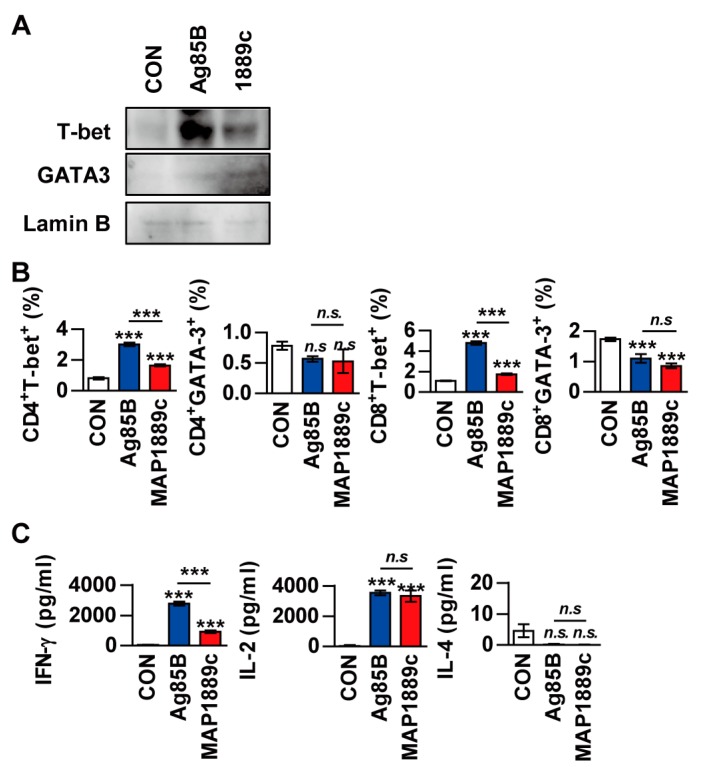
The frequencies of T-bet^+^ T cells and IFN-γ production are suppressed in restimulated splenocytes from MAP1889c-immunized mice. (**A**) Splenocytes isolated from Ag85B- and MAP1889c-immunized mice were stimulated with Ag85B and MAP1889c, respectively. After 72 h, the expression levels of T-bet and GATA-3 in T cells were assessed by immunoblotting. One representative blot out of five independent experiments is shown. (**B**) The frequency of GATA-3^+^ or T-bet^+^ T cells in splenocytes stimulated specifically with each Ag were analyzed by flow cytometry, and (**C**) production of IFN-γ, IL-2, and IL-4 in culture supernatants was measured by ELISA. Gating strategies are shown in Appendix A. All data are expressed as the mean ± SD (*n* = 3). *** *p* < 0.001 for the treatment compared with untreated controls (CON) or for the difference between treatment data. *n.s*., no significant difference.

**Figure 8 cells-09-00944-f008:**
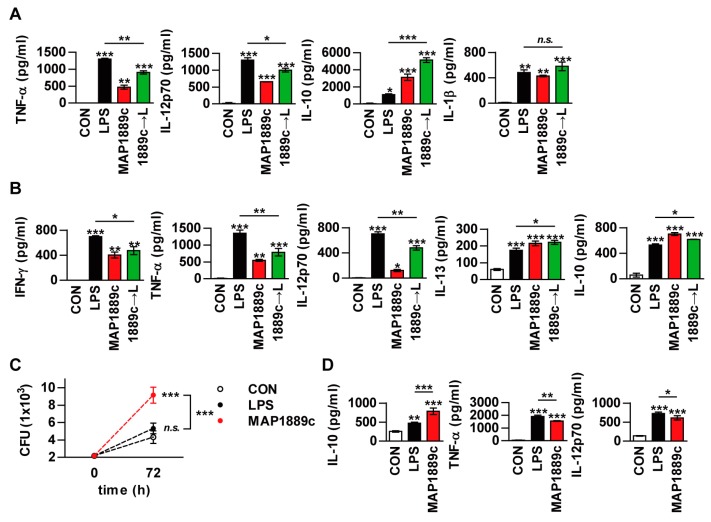
MAP1889c suppresses the LPS-induced pro-inflammatory response and promotes intracellular bacterial survival. (**A**) BMDCs were incubated with LPS (100 ng/mL) with or without pretreatment with MAP1889c (10 μg/mL) for 1 h. After 24 h, TNF-α, IL-12p70, IL-10, and IL-1β productions were analyzed by ELISA in culture supernatants. CON, medium control. (**B**) Unstimulated (CON)-, LPS-, MAP1889c-, and LPS-stimulated DCs pretreated with MAP1889c were co-cultured with splenocytes of naïve mice for 72 h at a DC to splenocytes ratio of 1:10. The cytokine levels in the culture supernatants were measured by ELISA. All data are expressed as the mean ± SD (*n* = 3). * *p* < 0.05, ** *p* < 0.01, and *** *p* < 0.001 for the treatment compared with untreated controls (CON) or for the difference between treatment data. (**C**,**D**) Bone marrow-derived macrophages (BMDMs) were infected with *Mycobacterium avium* at a multiplicity of infection (MOI) of 1 for 4 h and then further treated with amikacin to kill extracellular bacteria for 2 h, washed three times, and incubated with or without 10 μg/mL MAP1889c, or 100 ng/mL LPS for 72 h. Intracellular bacterial growth was determined by plating the cell lysates on 7H10 agar for 0 to 72 h (**C**). The amounts of IL-10, TNF-α, and IL-12p70 in the culture medium were measured by ELISA (**D**). The mean ± SD is shown for three independent experiments. * *p* < 0.05, ** *p* < 0.01, and *** *p* < 0.001 for the treatment compared with untreated controls (CON) or for the difference between treatment data. *n.s.*, no significant difference.

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
