# Peer review of "Mycobacterium avium subsp. paratuberculosis MAP1889c Protein Induces Maturation of Dendritic Cells and Drives Th2-Biased Immune Responses"

_cells, 2020, doi:10.3390/cells9040944_

Round 1

Reviewer 1 Report

This paper is overall well done. The work is extensive, critical controls are included and it is well written.  But a few things to address:

1) Figures 6E and 7B depict flow data.  The representative flow plots need to be shown for these data, either in the figure or in supplemental.  In particular, the FMO or isotype controls for the transcription factor staining needs to be shown- these are notoriously challenging stains.

2) In figure 8C and 8D, the authors investigate the effect of MAP1889c on bactericidal activity in BMDM.  The authors infect the BMDM, then measure bacterial burden 72 hours later.  I'm a bit surprised that the authors were able to detect differences in CFU at 72 hours- this bug is notoriously slow growing. I am also not convinced that the impaired bactericidal activity can be attributed to MAP1889C or IL-10.  It would be nice to see a dose response effect of MAP1889C, since this protein is already present due to the MAP infection.  Some measurement of the IL-10 induced by MAP infection would also be helpful, so inclusion of uninfected/LPS and uninfected MAP1889C treated MDM would also be useful. What about the use of IL-10 deficient BMDM?  Overall, this last experiment felt like an afterthought. I think that if the authors wish to include these data, they need to perform some additional experiments to support their conclusion on line 458. 

Minor typos to note: the sentence on lines 55-56 (Immature DCs are....) needs a bit of editing.  Line 58 - should be spleen OR adjacent lymph nodes.

Author Response

1) Figures 6E and 7B depict flow data.  The representative flow plots need to be shown for these data, either in the figure or in supplemental.  In particular, the FMO or isotype controls for the transcription factor staining needs to be shown- these are notoriously challenging stains.

2) In figure 8C and 8D, the authors investigate the effect of MAP1889c on bactericidal activity in BMDM.  The authors infect the BMDM, then measure bacterial burden 72 hours later.  I'm a bit surprised that the authors were able to detect differences in CFU at 72 hours- this bug is notoriously slow growing. I am also not convinced that the impaired bactericidal activity can be attributed to MAP1889C or IL-10.  It would be nice to see a dose response effect of MAP1889C, since this protein is already present due to the MAP infection.  Some measurement of the IL-10 induced by MAP infection would also be helpful, so inclusion of uninfected/LPS and uninfected MAP1889C treated MDM would also be useful. What about the use of IL-10 deficient BMDM?  Overall, this last experiment felt like an afterthought. I think that if the authors wish to include these data, they need to perform some additional experiments to support their conclusion on line 458. 

Minor typos to note: the sentence on lines 55-56 (Immature DCs are....) needs a bit of editing.  Line 58 - should be spleen OR adjacent lymph nodes.

All answers are in the attachment. Please see the attachment.

Reviewer 2 Report

Dr. Park et al. submitted their manuscript entitled “Mycobacterium avium subsp. paratuberculosis MAP1889c protein induces maturation of dendritic cells and drives Th2-biased immune responses” to the Cells. They paid their main attention to MAP1889c protein and induction of IL-10 as important anti-inflammatory/regulatory Th2 cytokine. The authors suppose that TLR4 signaling participates in the induction of IL-10 and other inflammatory cytokines.

The manuscript is clearly and precisely written, and methods are almost satisfyingly described, results clearly presented, and discussion is supported by results and literature references.

However, I have several objections:

L93: It is not clear what it means 1% antibiotics. Which antibiotics were used, and which was their concentrations?

L94: "1 mm sodium pyruvate". Please, correct the concentration.

L98: Which was the concentration of antibiotics?

L113: Please add a town and country for BD.

L237: The data in the graphs ARE NOT expressed as the mean values ± SD but as the mean value + SD. The same is in your supplement. Please, correct it in the Methods and graphs or their descriptions through the whole manuscript.

L255-259, L300-301, L330-332, L342-344, and L379: The texts belong to the Introduction or Discussion, but not to Results. Please, remove the texts or move them from the Results.

In your graphs, zero cytokine values are repeatedly depicted in the case of controls. What does it mean? Please, define sensitivities of your used cytokine assays in the Methods.

Author Response

First of all, we would like to thank the reviewers for their detailed and insightful review of our manuscript. Summarized comments below are italicized, followed by our replies in blue colored text. The modified part within the revised manuscript is shown in red color.

Comments and suggestions for Authors: 

Reviewer #2: 

Dr. Park et al. submitted their manuscript entitled “Mycobacterium avium subsp. Paratuberculosis MAP1889c protein induces maturation of dendritic cells and drives Th2-biased immune responses” to the Cells. They paid their main attention to MAP1889c protein and induction of IL-10 as important anti-inflammatory/regulatory Th2 cytokine. The authors suppose that TLR4 signaling participates in the induction of IL-10 and other inflammatory cytokines.

The manuscript is clearly and precisely written, and methods are almost satisfyingly described, results clearly presented, and discussion is supported by results and literature references.

(Answer) Thank you very much for your time and effort put into reviewing our manuscript.

However, I have several objections:

L93: It is not clear what it means 1% antibiotics. Which antibiotics were used, and which was their concentrations?

L98: Which was the concentration of antibiotics?

(Answer) Thank you for your kind advice. The 100 unit/ml of penicillin/streptomycin was used. We have clearly described the concentration and kinds of antibiotics used in the revised manuscript. The modified parts are shown in red colored font.

L94: "1 mm sodium pyruvate". Please, correct the concentration.

(Answer) Thank you for your kind indication. we have corrected it. The revised parts are shown in red colored font.

L113: Please add a town and country for BD.

(Answer) Thank you for your kind indication. We have added it. The revised parts are shown in red colored font.

L237: The data in the graphs ARE NOT expressed as the mean values ± SD but as the mean value + SD. The same is in your supplement. Please, correct it in the Methods and graphs or their descriptions through the whole manuscript.

(Answer) Thank you for mentioning the error of unexpected expression. We did statistical processing in mean values ± SD, but the error bar expression in the graph is expressed as only ‘above’, so you could think that it seems to be shown as + SD. Therefore, we have corrected all graphs in the revised manuscript.

L255-259, L300-301, L330-332, L342-344, and L379: The texts belong to the Introduction or Discussion, but not to Results. Please, remove the texts or move them from the Results.

(Answer) We are well aware that the text in the line mentioned is not the result. However, to make it easier for readers to read the paper, we have added a brief description at the beginning of the results. We have also seen many other papers expressing this. Therefore, we think it's natural to include these texts at the beginning of the results part.

  1. In your graphs, zero cytokine values are repeatedly depicted in the case of controls. What does it mean? Please, define sensitivities of your used cytokine assays in the Methods.

(Answer) We appreciate your comments. Zero values mean that cytokine levels are less than detectable range when calculated with standard curve for each cytokine. For example, in general, the lower cutoff values of IL-10, IL-12p70 and IFN-γ were 62.5 pg/ml, and those of TNF-α and IL-6 were 31.25 pg/ml. So an amount less than the lower cutoff value for each cytokine was considered as zero. 

Round 2

Reviewer 1 Report

Thank you for your responses.  I believe a few additional modifications are still required.

1) Thank you for including the gating strategies in the supplemental figures.  However, in your response, you indicated that you had included the FMO staining controls. It is not clear if the flow plots shown in the supplemental data are FMO controls?  If so, we need to see a representative stained sample compared to a FMO control stained sample, to determine how well the flow is performing.  Please revise the supplemental figures to include the necessary controls.

Author Response

All answers are in the attachment. Please see the attachment.

Reviewer 2 Report

Dear authors,

Thank you very much for your reply and that the changes were clearly labeled.

(Answer) Thank you for your kind advice. The 100 unit/ml of penicillin/streptomycin was used. We have clearly described the concentration and kinds of antibiotics used in the revised manuscript. The modified parts are shown in the red-colored font.

The streptomycin concentration is incorrectly in units/ml. Please, be a more observant and correct expression in both used antibiotics.

 (Answer) Thank you for mentioning the error of unexpected expression. We did statistical processing in mean values ± SD, but the error bar expression in the graph is expressed as only 'above', so you could think that it seems to be shown as + SD. Therefore, we have corrected all graphs in the revised manuscript.

I understood what you meant, of course.

You have three possibilities of how to depict SD (or SEM).

mean values ± SD: SD is depicted in BOTH SIDES of the mean value

or

mean values + SD: SD is ABOVE the mean value

or

mean values - SD:  SD is UNDER the mean value

in the first version of your manuscript, you combined the possibilities 1 (text) and 2 (graph). Now it is correct.

The mistakes in the depiction of means and SDs (SEMs) are widespread and usually tolerated in journals. It doesn't mean that it is correct.

(Answer) We are well aware that the text in the line mentioned is not the result. However, to make it easier for readers to read the paper, we have added a brief description at the beginning of the results. We have also seen many other papers expressing this. Therefore, we think it's natural to include these texts at the beginning of the results part.

I understand your opinion. Unfortunately, I do not agree with it.  I am very sorry. Please, see my original comment.

(Answer) We appreciate your comments. Zero values mean that cytokine levels are less than detectable range when calculated with standard curve for each cytokine. For example, in general, the lower cutoff values of IL-10, IL-12p70 and IFN-γ were 62.5 pg/ml, and those of TNF-α and IL-6 were 31.25 pg/ml. So an amount less than the lower cutoff value for each cytokine was considered as zero. 

Many thanks for your explanation. I know the principles of quantitative immunoassays. The absence of relevant information was the reason for my objection. I miss the detection limits in your revised manuscript again. Please, add them.

Author Response

First of all, we would like to thank the reviewers for their detailed and insightful review of our manuscript. Summarized comments below are italicized, followed by our replies in blue colored text. The modified part within the revised manuscript is shown in red color.

Comments and suggestions for Authors: 

Reviewer #2: 

Thank you very much for your reply and that the changes were clearly labeled.

1) The streptomycin concentration is incorrectly in units/ml. Please, be a more observant and correct expression in both used antibiotics.

(Answer) Thank you for your kind advice. We used as 100 unit/ml of penicillin/100 μg/ml of streptomycin. We have clearly described the concentration and kinds of antibiotics used in the 2nd revised manuscript. The modified parts are shown in red colored font.

2) The mistakes in the depiction of means and SDs (SEMs) are widespread and usually tolerated in journals. It doesn't mean that it is correct.

(Answer) Thank you for mentioning what we didn’t think of. 

3) I understand your opinion. Unfortunately, I do not agree with it.  I am very sorry. Please, see my original comment.

(Answer) Thank you very much for your time and effort put into reviewing our manuscript. We respect your opinion. For that reason, we have deleted and corrected the parts you mentioned to reflecting opinion on your original comments and also adjusted all references number.

4) Many thanks for your explanation. I know the principles of quantitative immunoassays. The absence of relevant information was the reason for my objection. I miss the detection limits in your revised manuscript again. Please, add them.

(Answer) We appreciate your comments. We understood that we were only writing in reviewer comments. In response to your comments, they were added in the section for ‘2.8. ELISA for cytokines’ of Material and Method as follows;

Amount less than the lower cutoff value for each cytokine was considered as zero. Calculating the standard curve for each cytokine, cutoff value of IL-10, IL-12p70 and IFN-γ were 62.5 pg/ml, and cutoff value of TNF-α and IL-6 were 31.25 pg/ml.